# Imputation-free Learning of Tabular Data with Missing Values using Incremental Feature Partitions in Transformer

## Abstract

Tabular data with varying missing values are imputed using an arbitrary imputation strategy for machine learning, which often compromises the data quality and reliability of data-driven outcomes. This article proposes imputation-free incremental attention learning (IFIAL) to streamline tabular data in a transformer without requiring initialization, imputation, or complete representation of missing values. A pair of attention masks are derived and retrofitted to the transformer, which incrementally learns small partitions of overlapping feature sets to enhance the efficiency and performance of learning representations. The average classification performance rank across 17 diverse tabular data sets shows the superiority of IFIAL over 11 state-of-the-art learning methods with or without missing value imputations. Further experiments substantiate the robustness of IFIAL against the varying types and rates of missing values. The proposed method is one of the first solutions to enable deep attention learning of tabular data without requiring missing-value imputation or learning a complete data representation for classification. The source code for this paper is available at [1].

## 1 Introduction

Missing values at varying rates and types are ubiquitous in countless application domains, especially in electronic health records (EHR). The handling of missing values remains critical for data quality and data-driven outcomes. Unfortunately, state-of-the-art machine and deep learning (DL) methods are not designed to learn directly from tabular data with missing values. Therefore, a complete data representation is obtained to enable the learning of tabular data using one of the three strategies: 1) excluding missing samples and features, 2) imputing missing entries with model-generated synthetic values, and 3) end-to-end learning of a complete data representation after initializing the missing values. Synthetic values generated by the model inevitably alter feature importance, data statistics and correlations between the features, dampening the predictive ability of the data Arifeen & Petrovski (2022). It is not trivial to identify the statistical type of missing values to select the optimal imputation strategy. Therefore, there is no best imputation method for all data sets or missing value types Payrovnaziri et al. (2021); Ribeiro & Freitas (2021); Samad et al. (2022). This paper aims to bypass the requirement and limitations of imputing missing values before learning representations and address data quality concerns by proposing a new Imputation-free Incremental Attention Learning (IFIAL) strategy. IFIAL engineers a set of attention masks in a transformer to directly learn tabular data in small feature partitions without initializing or imputing missing values.

The remainder of this paper is organized as follows. Section 2 reviews the relevant literature on incremental learning and representation learning of tabular data with missing values to motivate the contributions of our work in Section 3. Section 4 introduces the proposed NAIL strategy, integrating attention mask engineering and a new incremental learning strategy. Section 5 discusses the experimental setup, data sets, missingness scenarios and baselines, and the evaluation method. Section 6 compares the performance of IFIAL with the state-of-the-art baselines, including robustness and ablation studies. Section 7 summarizes the key findings, provides insight into the results, and highlights the limitations of the proposed method. Finally, Section 8 concludes the paper.

---

[1] https://anonymous.4open.science/r/ifial-435B

## 2    RELATED WORK

Feature incremental learning (FIL) is one of the cornerstones of deep learning (DL), which is not viable in traditional ML. Although highly successful with homogeneous image features Li et al. (2022), incremental learning is challenging in the heterogeneous feature space of tabular data Liu et al. (2023). Moreover, the superiority of traditional ML in tabular data over deep learning is well documented in the literature Borisov et al. (2022). Some recent work has achieved FIL on tabular data Liu et al. (2023); Ahamed & Cheng (2024); Kim et al. (2024); Wang & Sun (2024). A common approach to FIL trains multiple models separately to accommodate new features and aggregate model predictions through a weighting mechanism Kim et al. (2024); Liu et al. (2023). Alternatively, mamba Ahamed & Cheng (2024) and the transformer-based method Wang & Sun (2024) use a single model to train on additional features incrementally as they become available. However, the efficacy of FIL has not been investigated in learning tabular data with missing values, especially in handling missing values. The state-of-the-art methods for handling missing values are broadly categorized into traditional and deep imputations of missing values before learning representations. Traditional methods, such as MICE Beesley et al. (2021), missForest Stekhoven & Bühlmann (2011), and MLP in multiple imputations Samad et al. (2022)), use regression models to estimate and impute missing values from the relationship between features. Deep imputation methods include Generative Adversarial Imputation Nets (GAIN) Yoon et al. (2018), diffusion-based imputation (Diffputer) Zhang et al. (2025), variational autoencoders Hong & Hao (2023), and denoising autoencoders Shang et al. (2017). Sun et al. Sun et al. (2023) report that GAIN has limited performance when missing values are not at random (MNAR), where MICE often outperforms GAIN. However, generative adversarial networks (GANs) can be unstable during optimization, particularly in terms of model convergence Jarrett et al. (2022). Diffusion models use a simplified assumption about feature distributions (e.g., a Gaussian distribution), which may not be accurate for all features. Chen et al. Chen et al. (2024) report the superiority of GAIN over its diffusion counterparts, such as MissDiff Ouyang et al. (2023) and CSDI_T Zheng & Charoenphakdee (2022). These data-centric assumptions and limitations of the imputation methods explain why no single imputation method is ideal for all data sets and missing-value types. Many researchers are reluctant to use DL imputation methods to prepare data due to high computational costs, lack of trust, and interpretability Bansal et al. (2021); Kamal et al. (2020). In fact, several DL methods have been proposed to directly learn data with missing values without using a standalone imputation step Le Morvan et al. (2020). Similar methods first initialize missing values to subsequently learn a complete data representation during an end-to-end supervised learning task, such as classification and regression. Although the contribution of missing values to learning the complete data representation is masked out, these strategies do not enable a classifier to directly learn from data with missing values. Moreover, an observed zero value is not distinguished from zeros representing missing entries in a feature. On the other hand, decision tree-based ML models such as Random Forest and XGBoost Chen & Guestrin (2016) use "Missing Incorporated in Attributes" (MIA)Twala et al. (2008) to directly learn data with missing values. However, traditional ML methods can be prone to overfitting high-dimensional feature spaces when incremental feature learning is not viable.

## 3    MOTIVATION AND CONTRIBUTIONS

A standalone data imputation method can compromise data quality when statistics of missing value type are unknown. Moreover, tabular data with mixed feature types require separate classifier and regressor models to impute numerical and categorical features, respectively. When the entire data set is imputed using observed values, similar to Du et al. (2024); Lall & Robinson (2022), the test data fold in the subsequent classification is likely to contain information leaked from the training data folds. Alternatively, the joint imputation and classification methods He et al. (2024) impute data (X) to fit the classification targets (y), as X←f(y), in contrast to a classification task, y ← f(X). Therefore, joint imputation and classification can leak information from class labels to data values (X). On the other hand, without complete data representation, tabular data would consist of features with varying missing rates, which can impact representation learning based on between-feature relationships. In this context, sequential learning of data in small feature partitions, starting from the partition with the lowest missing rates, can mitigate the effect of varying missing rates and improve computational efficiency.

This paper makes four key contributions to address the challenges in learning tabular data with missing values. First, we present one of the first studies to streamline tabular data with missing values in a deep attention learning framework without requiring imputation or even initialization of missing values. Second, the proposed method derives and retrofits two attention masks to a feature-tokenized transformer to exclude missing entries from attention scoring. Third, we propose a new incremental learning strategy to mitigate the effect of varying missing rates in the feature space and allow efficient attention learning of O ($n^2$), which otherwise results in an out-of-memory problem with large feature spaces Rabbani et al. (2024). Fourth, the entire deep attention framework is designed to be computationally efficient and performance-wise superior to state-of-the-art baselines, both with and without missing value imputations.

# 4 METHODOLOGY

This section presents the new attention mask design and the incremental feature learning strategy used to develop the proposed IFIAL method for learning tabular data with missing values.

## 4.1 PRELIMINARIES

A tabular data set $X \in \Re^{n \times d} = \{X_{obs}, X_{miss}\}$ with $r\%$ missing values is composed of observed ($X_{obs}$) and missing ($X_{miss}$) values. An imputation model first initializes $X_{miss}$ using mean or median values as ($\hat{X}_{miss}$). The model then iteratively refines the initial estimates of missing values as $\hat{X}_{miss}^{i+1}$ using $X_{obs}$ and $\hat{X}_{miss}^{i}$ estimates from the previous iteration. An imputation model makes the assumption that $X_{obs}$ is predictive of $X_{miss}$, which may not always be true depending on the predictive model, data statistics, and missing value types. This assumption can be entirely waived when a data set can be learned using $X_{obs}$ alone without estimating the $X_{miss}$ portion. A data set with missing values can be learned through FIL using $X_{obs}$ alone. FIL updates an ML model when additional features become available in a multiphase data collection scenario. An ML model trained on the data set $X_1 \in \Re^{N_1 \times d_1}$ can be updated by a data set collected in the second phase $X_2 \in \Re^{N_2 \times (d_1 + d_2)}$ with additional $d_2$ features, and so on. Without FIL, the conventional approach combines the two tabular data sets $X_1$ and $X_2$ to train the ML model from scratch, where samples of $X_1$ would have values missing for additional $d_2$ features. Unfortunately, traditional ML is not usually designed to learn directly from data with missing values. In this context, we repurpose FIL to learn data with missing values directly, eliminating the need for data imputation or exclusion, as discussed in subsequent sections.

## 4.2 PROPOSED IFIAL ALGORITHM

The proposed Imputation-free Incremental Attention Learning (IFIAL) completely avoids imputation or initialization of missing values $X_{miss}$. Suppose a labeled tabular data set with missing values $D = \{X, y\}$ has $n$ samples and $d$ features. First, we sort the features $f_1, f_2, \ldots, f_m$ in ascending order of missing rates. The sorted features are partitioned into subsets of $k$ features with $s = ceil(\frac{k}{2})$ overlaps, as shown in Figure 1. The overlap between two consecutive feature subsets facilitates incremental learning using $P$ partitions of features, as follows.

$$P = 1 + \lceil \frac{d - k}{k - \lceil k/2 \rceil} \rceil \tag{1}$$

These $P$ partitions are used to achieve incremental learning using a feature-tokenized transformer (FTT) Wang & Sun (2024); Huang et al. (2020); Schambach et al. (2023) as follows. An FTT classifier model ($FTT_1$) is trained using the first partition $P_1 = X\{f_1, ..., f_k\}$ of $k$ features. The next incremental learning session uses the second partition $P_2 = X\{f_s, ..., f_{k+s}\}$ to train the previous $FTT_1$ model to $FTT_2$. Incremental learning continues until the $FTT_P$ model is obtained using the last feature partition, as illustrated in Figure 1. The trained $FTT_P$ model directly classifies a separate test data fold with missing values. Notably, FTT tokenizes features and uses a pre-trained language model to obtain embeddings for feature names and categorical features.In contrast, numerical values are passed through a linear projection layer to obtain corresponding embeddings. The tabular data representation is initialized by aggregating the embeddings of feature names and numerical and categorical feature values, which are subsequently fine-tuned for classification using a gated

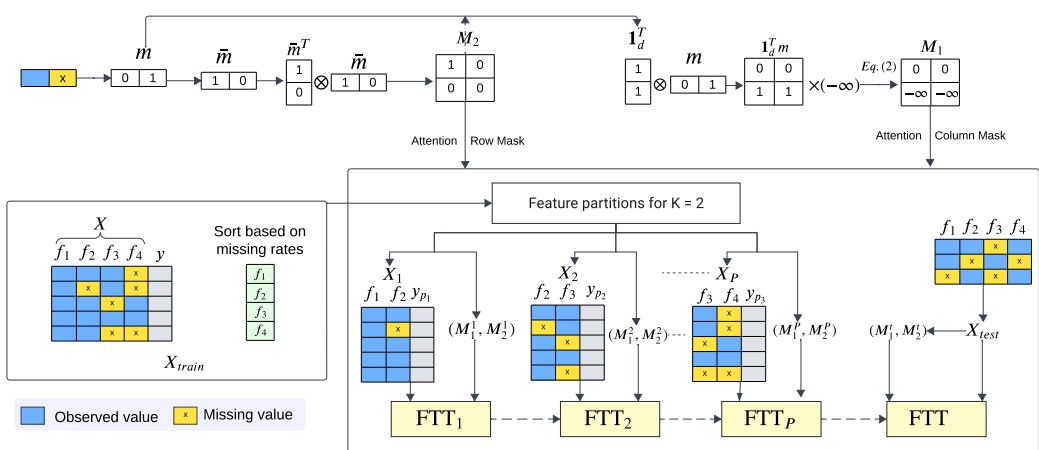

Figure 1: Imputation-free Incremental Attention Learning (IFIAL) method uses $P$ fixed-sized overlapping feature partitions to incrementally train a Feature-Tokenized Transformer (FTT). Attention masks: $M_1$ operates as the attention column mask and $M_2$ as the attention row mask to filter missing feature values from attention scoring.

---

**Algorithm 1** THE IFIAL ALGORITHM

---

**Input:** Tabular data with missing values, $\{X \in \Re^{N \times d}, y\}$
**Parameters:** $k$: number of features in partitions.
**Output:** Trained classifier model, $\mathcal{M}(\theta)$
**Procedure:**
$\{X^{obs}, X^{miss}\} \leftarrow \{X, y\}$
Missing rate of $f_j$ feature, $r_j \leftarrow \frac{length(X^{miss}(:,j))}{length(X(:,j))}$
$\{f_1, f_2, ..., f_d\} \leftarrow \{r_1 < r_2 < ... < r_d\}$, sorting $f_i$.
$\{P_1, P_2, ...\} \leftarrow \{X_1(f_1, f_2), X_2(f_2, f_3), ...\}$, when k=2.
**for** $P_i \in \{P_1, P_2, ...\}$ **do**
  $\{X_i^{obs}, X_i^{miss}\} \leftarrow P_i$
  $m \leftarrow X_i^{\text{miss}}$, Missing mask vector
  $M_1 \leftarrow m$, Eq. 2
  $M_2 \leftarrow \bar{m}^T \bar{m}$
  $\mathcal{M}(\theta) \leftarrow \{(X_i, M_1, M_2), y_i\}$, Eq. 5
**end for**

---

transformer. The proposed IFIAL algorithm is presented in Algorithm 1. Missing values ($X^{miss}$) of X are excluded from attention scoring using attention masks designed as follows.

A binary vector $m \in \Re^{1 \times d}$ uses 1s to identify features with missing values and 0s otherwise. An outer product between a column vector of 1s and $m$ yields a $(d, d)$ binary matrix ($M_1$) with columns of 1s corresponding to the features with missing values. These columns of 1s are replaced with large negative values in Equation 2. An exponential function transforms large negative values in $M_1$ representing missing features to zeros, as shown in Equation 3.

$$M_1 = -\infty \odot (\mathbf{1}_d^T m) \tag{2}$$

$$exp(M_1) = \begin{cases} 0, & \text{if } M_1(:) = -\infty \\ 1, & \text{if } M_1(:) = 0 \end{cases} \tag{3}$$

$M_1$ filters the attention columns corresponding to the features with missing values using the softmax function in Equation 4. The attention rows corresponding to features with missing values are eliminated using another binary mask, $M_2$. Here, $M_2 = \bar{m}^T \bar{m}$ obtained from the complement of $m$

($\bar{m}$) includes 0s in rows and columns that correspond to missing features and 1s otherwise.

$$\text{Attention} \quad = \quad \text{softmax}(\frac{QK^T}{\sqrt{h}} + M_1) \odot M_2 \tag{4}$$

$$= \quad (\frac{\exp{(\frac{QK^T}{\sqrt{h}} + M_1)}}{\sum \exp{(\frac{QK^T}{\sqrt{h}} + M_1)}} \odot M_2)$$

$$= \quad \frac{\exp{(\frac{QK^T}{\sqrt{h}})} * \exp{(M_1)}}{\sum \exp{(\frac{QK^T}{\sqrt{h}})} * \exp{(M_1)}} \odot M_2 \tag{5}$$

Combining Equations 3 and 5, a $d \times h$ attention head in Equation 6 yields a $h$ dimensional embedding for each of the $d$ features. The query ($Q$), key ($K$), and value ($V$) matrices obtain the attention head in Equation 6.

$$\text{Head} = \begin{cases} (0 * I_{d \times h}) * V, & \text{if } M_1(:) = \text{-}\infty, M_2(:) = 0 \\ \text{softmax}(\frac{QK^T}{\sqrt{h}})V, & \text{if } M_1(:) = 0, M_2(:) = 1 \end{cases} \tag{6}$$

$M_1$ uses $-\infty$ to mask the attention columns with missing features. The remaining rows of attention with missing features are masked by 0s of $M_2$. Otherwise, $M_1$ and $M_2$ retain the attention scores of all observed feature pairs using 0s and 1s, respectively.

## 5 EXPERIMENTS

We perform experiments on three types of missing values: 1) missing completely at random (MCAR), 2) missing not at random (MNAR), and 3) missing values that occur naturally in data sets. Here, MCAR and MNAR types are simulated at a missing rate ranging from 10% to 50% at 10% intervals.

### 5.1 TABULAR DATA SETS

Table 4 in Appendix A.1 summarizes the tabular data sets used in this article. For better generalizability, we have selected 17 diverse data sets of varying sizes and a mix of numerical and categorical features from the OpenML repository Bischl et al. (2021). Tabular data sets with large sample sizes (>10,000) are often selected to facilitate deep learning Caruso et al. (2024); Gorishniy et al. (2021). However, most tabular data sets in practice have relatively small sample sizes (<1000) Rabbani et al. (2025), which are more challenging data sets for deep learning methods. Therefore, the sample sizes of these data sets range from 155 to 72983, while the feature dimension ranges from 7 to 39. The first 13 data sets are available in complete form and are used to simulate MCAR and MNAR-type missing values at varying rates. The other four data sets with IDs 55, 6332, 41162, and 41440 have natural missing values at the rate of 48.8%, 48.7%, 30.2%, and 20.6%, respectively.

### 5.2 MODEL IMPLEMENTATION AND EVALUATION

The FTT classifier, adopted from Wang & Sun (2024), consists of a transformer with two encoder layers. The encoder layers transform numerical and categorical features into 128-dimensional embeddings. Each transformer layer includes eight attention heads and a feedforward network with a 2048-dimensional hidden layer, ReLU activation functions, and a dropout rate of 0.3. We have used the Adam optimizer with a fixed learning rate of $1 \times 10^{-5}$ and no weight decay. The model is trained for up to 300 epochs using a batch size of 128. Early stopping is used with a patience of 50 epochs, terminating training when the validation loss ceases to improve for 50 consecutive epochs.

A five-fold cross-validation results in the average area under the receiver operating characteristic curves (AUC). The average rank of each method is reported based on AUC scores Borisov et al. (2022) for individual missing value types. Similarly to Tokar & Sanner (2024), we use a win matrix to show the percentage of experimental scenarios in which one method outperforms the other. Furthermore, the resilience of a method to varying missing rates is evaluated by comparing the corresponding AUC scores with the reference AUC score of the complete data. We primarily evaluate the proposed incremental learning method for three partition sizes, k = 2, 3, and 4. Our analysis in Figure 3 in

Table 1: Average performance rank of the proposed (IFIAL) and baselines for 13 data sets with MCAR type missing values.

| Method | Imputation | 10% | 20% | 30% | 40% | 50% | Avg. Rank | Overall Rank |
|---|---|---|---|---|---|---|---|---|
| Median-GBT | Yes | 7.8(0.05) | 7.4(0.05) | 7.5(0.04) | 8.3(0.06) | 7.2(0.05) | 7.7(0.05) | 7 |
| Median-FTT | Yes | 8.7(0.07) | 8.7(0.07) | 8.9(0.06) | 9.0(0.06) | 9.8(0.07) | 9.0(0.07) | 12 |
| MICE-GBT | Yes | 6.5(0.05) | 6.8(0.05) | 8.3(0.05) | 8.2(0.05) | 7.8(0.06) | 7.5(0.05) | 6 |
| MICE-FTT | Yes | 8.0(0.05) | 7.6(0.05) | 6.9(0.04) | 6.8(0.04) | 7.2(0.05) | 7.3(0.05) | 5 |
| GAIN-GBT | Yes | 10.0(0.05) | 9.2(0.05) | 7.4(0.05) | 7.1(0.06) | 6.6(0.04) | 8.0(0.05) | 8 |
| GAIN-FTT | Yes | 7.8(0.06) | 6.6(0.06) | 6.8(0.06) | 7.1(0.07) | 6.8(0.06) | 7.0(0.06) | 4 |
| Diffputer-GBT | Yes | 9.1(0.06) | 9.1(0.06) | 10.2(0.06) | 11.9(0.06) | 9.7(0.06) | 10.0(0.06) | 14 |
| Diffputer-FTT | Yes | 8.7(0.06) | 9.7(0.06) | 10.0(0.06) | 9.3(0.06) | 10.1(0.06) | 9.5(0.06) | 13 |
| MIA-Xgboost | No | 9.1(0.05) | 8.9(0.05) | 8.4(0.05) | 8.9(0.05) | 8.2(0.05) | 8.7(0.05) | 11 |
| MIA-LightGBM | No | 8.5(0.04) | 9.2(0.04) | 9.0(0.05) | 9.5(0.05) | 7.2(0.05) | 8.7(0.05) | 10 |
| AM-FTT | No | 9.3(0.06) | 10.5(0.06) | 10.8(0.06) | 10.3(0.07) | 11.5(0.07) | 10.5(0.06) | 15 |
| IFIAL (K =2) | No | 8.2(0.04) | 8.2(0.05) | 7.6(0.04) | 7.8(0.05) | 8.5(0.06) | 8.0(0.05) | 9 |
| IFIAL (K =3) | No | 6.7(0.05) | 6.0(0.04) | 5.9(0.04) | 5.9(0.04) | 6.2(0.05) | 6.1(0.05) | 2 |
| IFIAL (K =4) | No | 6.4(0.04) | 6.0(0.04) | 6.5(0.05) | 5.2(0.04) | 7.0(0.05) | 6.2(0.05) | 3 |
| IFIAL (K =d/2) | No | 5.5(0.05) | 6.1(0.05) | 5.8(0.05) | 4.8(0.04) | 6.2(0.05) | 5.7(0.05) | 1 |

Appendix A.2 suggests that incremental learning using feature partitions is computationally more efficient than learning the entire feature space ($d$ features) when k is less than $\lceil d/2 \rceil$. Therefore, the performance of our model is reported for partitions when $k = \frac{d}{2}$.

## 5.3 BASELINE METHODS

The baseline methods are of two types: one type requires imputation before classification, and the other can directly classify data without a standalone imputation. The imputation requirement before classification is satisfied by median value imputation, MICE Beesley et al. (2021), GAN-based imputation method GAIN Yoon et al. (2018), and diffusion-based imputation method, Diffputer Zhang et al. (2025). Unlike mean values, median values are robust to outliers and are recommended for low missing rates. The subsequent classification of imputed data sets is achieved using two state-of-the-art models for tabular data, gradient boosting trees (GBTs), and FTT Wang & Sun (2024). On the other hand, XGBoost Chen & Guestrin (2016) and LightGBM Ke et al. (2017) can use the Missing Incorporated in Attributes (MIA) strategy Twala et al. (2008), as mentioned in Section 2, to directly classify data with missing values. XGBoost and LightGBM treat missing values as distinct categories during decision tree splits. The FTT classifier can filter missing values using an attention mask (AM) presented in Equation 4. Therefore, an AM-FTT framework, similar to Caruso et al. (2024), is considered a baseline. The 11 baseline methods are 1) median - GBT, 2) median - FTT, 3) MICE - GBT, 4) MICE - FTT, 5) GAIN - GBT, 6) GAIN - FTT, 7) Diffputer - GBT, 8) Diffputer - FTT, 9) AM - FTT, 10) MIA - XGBoost, and 11) MIA - LightGBM.

## 6 RESULTS

All results are obtained using an Ubuntu 22.04 machine with an Intel(R) Xeon(R) W-2265 CPU (24 logical cores) running at 3.70GHz, 64GB of RAM, and a Quadro RTX A4000 GPU with 16GB of video memory. The performance of the proposed IFIAL method is compared with the baseline methods using 1) the average rank based on AUC scores, 2) the win matrix for the pairwise method comparison, and 3) the robustness of the performance to increasing missing rates. Appendix A.5 presents the AUC scores specific to data sets of type MCAR in Table 6 and of type MNAR in Table 7.

## 6.1 AVERAGE PERFORMANCE RANK

Table 1 compares the proposed IFIAL method with 11 baselines for MCAR-type data. In single value imputation, median imputation is better with the GBT classifier than FTT, regardless of missing rates. MICE outperforms the median value imputation. The GAN-based imputation method (GAIN) with the FTT classifier outperforms MICE imputation for all missing rates. In contrast, the diffusion-based imputation method performs the worst among all imputation techniques. Among the methods without imputation, the MIA-based models (MIA-XGBoost and MIA-LightGBM) outperform AM-FTT. The average performance rank shows that the proposed IFIAL (K = d / 2) outperforms the 11 baselines. IFIAL with other feature partition sizes, including IFIAL (K = 3) and IFIAL (K = 4), also rank among

Table 2: Average performance rank of the proposed (IFIAL) and baselines for 13 data sets with MNAR-type missing values.

| Method | Imputation | 10% | 20% | 30% | 40% | 50% | Avg. Rank | Overall Rank |
|---|---|---|---|---|---|---|---|---|
| Median-GBT | Yes | 7.2(0.05) | 7.3(0.06) | 7.3(0.06) | 8.5(0.05) | 5.8(0.06) | 7.2(0.06) | 6 |
| Median-FTT | Yes | 7.4(0.05) | 7.2(0.05) | 6.5(0.06) | 7.7(0.06) | 8.8(0.06) | 7.5(0.06) | 9 |
| MICE-GBT | Yes | 9.1(0.05) | 8.0(0.04) | 7.0(0.06) | 7.2(0.06) | 5.8(0.05) | 7.4(0.05) | 8 |
| MICE-FTT | Yes | 11.1(0.05) | 9.5(0.06) | 10.1(0.06) | 9.5(0.06) | 10.0(0.07) | 10.0(0.06) | 13 |
| GAIN-GBT | Yes | 8.5(0.04) | 8.8(0.05) | 9.9(0.04) | 8.6(0.04) | 8.8(0.04) | 8.9(0.04) | 12 |
| GAIN-FTT | Yes | 11.5(0.07) | 11.9(0.08) | 11.2(0.06) | 12.3(0.06) | 12.9(0.06) | 12.0(0.06) | 15 |
| Diffputer-GBT | Yes | 9.2(0.06) | 10.5(0.07) | 10.9(0.06) | 10.0(0.05) | 9.8(0.06) | 10.1(0.06) | 14 |
| Diffputer-FTT | Yes | 8.4(0.07) | 7.5(0.06) | 8.8(0.07) | 8.0(0.06) | 10.1(0.06) | 8.6(0.06) | 11 |
| MIA-Xgboost | No | 5.5(0.05) | 6.8(0.05) | 7.5(0.05) | 8.5(0.05) | 7.2(0.06) | 7.1(0.05) | 5 |
| MIA-LightGBM | No | 6.5(0.05) | 7.7(0.04) | 7.2(0.05) | 7.0(0.05) | 7.9(0.06) | 7.3(0.05) | 7 |
| AM-FTT | No | 9.9(0.04) | 7.5(0.06) | 8.2(0.05) | 8.9(0.05) | 8.2(0.06) | 8.6(0.06) | 10 |
| IFIAL (K =2) | No | 7.2(0.05) | 7.1(0.06) | 6.3(0.06) | 6.4(0.05) | 5.7(0.06) | 6.5(0.06) | 4 |
| IFIAL (K =3) | No | 6.6(0.05) | 6.2(0.06) | 6.8(0.07) | 5.9(0.05) | 6.2(0.06) | 6.3(0.06) | 2 |
| IFIAL (K =4) | No | 6.2(0.05) | 7.2(0.06) | 6.3(0.06) | 5.7(0.05) | 6.4(0.06) | 6.4(0.06) | 3 |
| IFIAL (K =d/2) | No | 5.8(0.04) | 6.7(0.07) | 6.1(0.06) | 5.7(0.06) | 6.4(0.06) | 6.1(0.06) | 1 |

Table 3: Mean AUC scores for four data sets with natural missing values: data set ID: 55 with 48.8%, ID: 6332 with 48.7%, ID: 41162 with 30.2%, and ID: 41440 with 20.6% missing values.

| ID | Median-GBT | Median-FTT | MICE-GBT | MICE-FTT | GAIN-GBT | GAIN-FTT | Diffputer-GBT | Diffputer-FTT | MIA-Xgboost | MIA-LightGBM | AM-FTT | IFIAL (K=d/2) |
|---|---|---|---|---|---|---|---|---|---|---|---|---|
| 55 | 0.806 (0.08) | 0.809 (0.04) | 0.790 (0.09) | 0.840 (0.04) | 0.779 (0.12) | 0.837 (0.05) | 0.788 (0.10) | 0.805 (0.06) | 0.757 (0.06) | 0.782 (0.07) | 0.806 (0.04) | 0.846 (0.04) |
| 6332 | 0.867 (0.02) | 0.743 (0.01) | 0.868 (0.02) | 0.796 (0.04) | 0.739 (0.07) | 0.808 (0.03) | 0.838 (0.03) | 0.781 (0.05) | 0.912 (0.01) | 0.757 (0.06) | 0.814 (0.03) | 0.861 (0.03) |
| 41162 | 0.737 (0.02) | 0.738 (0.01) | 0.738 (0.01) | 0.739 (0.01) | 0.737 (0.01) | 0.739 (0.01) | 0.740 (0.01) | 0.738 (0.01) | 0.736 (0.02) | 0.740 (0.01) | 0.739 (0.01) | 0.741 (0.01) |
| 41440 | 0.681 (0.01) | 0.654 (0.02) | 0.684 (0.01) | 0.671 (0.01) | 0.648 (0.03) | 0.649 (0.02) | 0.679 (0.01) | 0.653 (0.03) | 0.662 (0.01) | 0.679 (0.01) | 0.670 (0.01) | 0.675 (0.01) |
| Avg. Rank | 4.2 (0.03) | 7.5 (0.02) | 4.2 (0.04) | 5.0 (0.04) | 10.0 (0.05) | 5.2 (0.02) | 7.2 (0.04) | 7.0 (0.03) | 8.2 (0.02) | 6.2 (0.04) | 5.2 (0.02) | 2.5 (0.02) |

the top three. Even at low missing rates (10%–30%), IFIAL (K = d/2) maintains a strong performance rank, while at higher rates it shows clear superiority. These results confirm the robustness and consistent performance of the proposed IFIAL method for MCAR-type data.

Table 2 presents the average classification performance ranks for MNAR-type data. Imputation methods that present competitive performance for MCAR-type data (GAIN-FTT and MICE-FTT) show a substantial performance drop for the MNAR type, ranking 15th and 13th, respectively. In contrast, MIA-based models (MIA-XGBoost and MIA-LightGBM) perform better on MNAR data than MCAR data. This observation is consistent with previous findings Van Ness et al. (2023) suggesting the MIA method performs better when the missingness is informative, such as MNAR. At lower missing rates (10% and 20%), the average ranks suggest that MIA methods outperform all methods that require an imputation step. However, median-GBT consistently outperforms MIA and other imputation methods for a missing rate greater than 20%. On the other hand, IFIAL with other partition sizes (K = 2, 3 and 4) is among the top four best methods, while (K = d / 2) is the best performing method overall. These results suggest that the proposed IFIAL method is not affected by missing value types, unlike other imputation-based methods. Table 3 presents the performance of four data sets with natural missing values. The proposed IFIAL method (K = d/2) achieves the best average performance rank (2.5 (0.02)), while the second-best method is the median imputed data with a GBT classifier (average rank (4.2 (0.03)). Interestingly, MIA-Xgboost performs the best on a data set (ID 6332) with limited samples but relatively large feature space. For data sets with large sample sizes, the differences in model performance are not substantial, which does not pose a challenge in learning from small data sets.

## 6.2 Win matrix, computational time, and robustness

There are 65 experimental scenarios for each missing value type using 13 data sets and five missing value rates. Figure 2 (a) for the MCAR-type data shows that the proposed IFIAL method is superior to the baselines in more than 50% of the experimental scenarios. In the best case, IFIAL outperforms Diffputer-FTT in 82% of the scenarios. Figure 2 (b) for the MNAR-type data shows that IFIAL is always superior to the baselines for more than 50% of the experimental scenarios. In the best case, IFIAL outperforms GAIN-FTT in 83% of the scenarios, which is 55% for MCAR-type data. Table 5 in Appendix 5 presents the run time for the imputation and classification steps of the baseline methods. As expected, methods faster than IFIAL include GAIN-GBT, Diffputer-GBT, AM-FTT, MIA-Xgboost, and MIA-LightGBM, mainly due to the fast GBT classifier model. However, the

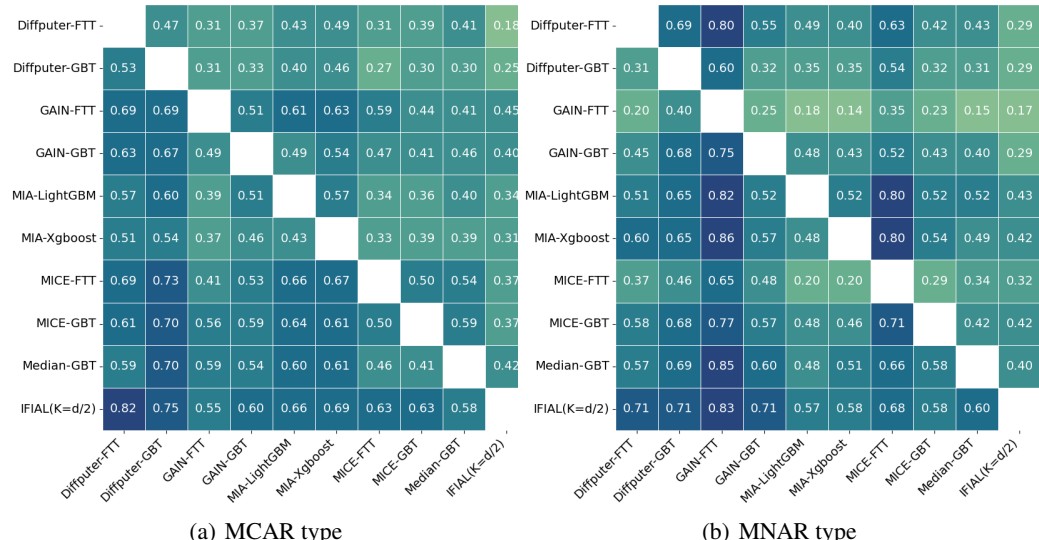

(a) MCAR type          (b) MNAR type

Figure 2: Win matrix. Values are the fraction of experimental scenarios in which the row methods outperform the methods in the columns.

classification performance rank of these methods is worse than that of IFIAL. One computational benefit of IFIAL is that it saves the time required for imputation. Appendix A.4, Figure 5 shows the effect of varying the missing rates on the performance of five top-performing methods. The reference AUC scores for the robustness analysis are obtained by classifying complete data sets with 0% missing values. The AUC score decreases relative to the reference AUC as the missing rate increases. Figure 5(a) for the MCAR-type data shows the robustness of the proposed method at missing rates higher than 30%. GAIN-FTT is the most robust method for up to a 20% missing rate. In contrast, Figure 5(b) for MNAR-type data shows that the robustness of the proposed IFIAL method to high missing rates is better than all other baselines.

## 6.3 ABLATION STUDIES

In an ablation study, we vary the feature partition size (k) for three representative data sets: Kc2 (ID 1063), Diabetes (ID 37), and Dresses sales (ID 23381). Figure 4 in Appendix A.3 shows the k values that yield the best AUC scores for individual data sets. In general, incremental learning of features is superior to learning with all features at once (that is, without incremental learning, k = d). The best average AUCs are obtained for the data sets Kc2 ($d = 21$) at k=10, Diabetes ($d = 8$) at k=6, and Dresses sales ($d = 12$) at k = 6. Therefore, the value of k can be chosen as half of the feature dimension for the best classification performance. This observation aligns with the computational cost analysis presented in Figure 3 in Appendix A.2, where $k < \frac{d}{2}$ is computationally more efficient than using the entire feature space ($d$). IFIAL outperforms all or most of the baselines at k = 3, which can be further improved by increasing the partition size to $\frac{d}{2}$.

## 7 DISCUSSIONS

The paper proposes incremental attention learning of tabular data as a novel mechanism to bypass the requirements of initializing, imputing, or completing the data. The findings of the paper can be summarized as follows. First, the proposed IFIAL method outperforms all baseline methods with and without data imputation, regardless of the missing value types. Second, state-of-the-art imputation methods are sensitive to missing value types, especially when missing values are not at random (MNAR). The proposed IFIAL method entirely avoids the imputation of missing values and, therefore, is a superior choice when the missing value type is unknown or complex. Third, while retrofitting a pair of attention masks enables imputation-free learning of tabular data representations, incremental learning of feature partitions is essential for overall efficiency and performance. In comparison, the performance of AM-FTT suggests that attention masks and the FTT model are insufficient to achieve

optimal performance. Fourth, the IFIAL method offers the best performance and computational time trade-off with better resilience to missing value rates than baselines. In general, our proposed method renders existing missing value imputation methods unnecessary from both computational efficiency and performance standpoints.

In contrast, the literature continues to propose novel methods for imputing missing values under varying constraints and assumptions. A recent survey reports that deep generative models (e.g., GAIN, VAE) require a large sample size ($n > 30,000$); otherwise, traditional and statistical imputation methods (e.g., MICE) perform better Sun et al. (2023). This may explain why MICE is superior to GAIN in our results for data with a decent sample size. The requirement of a large sample size does not apply to IFIAL. Furthermore, the recent diffusion-based method, Diffputer Zhang et al. (2025) provided a comprehensive benchmarking against multiple advanced baselines, including diffusion-based methods (MissDiff Ouyang et al. (2023), TabCSDI Zheng & Charoenphakdee (2022)), generative adversarial methods (GAIN Yoon et al. (2018), MIWAE Mattei & Frellsen (2019)), iterative imputation (MICEResche-Rigon & White (2018), EM, HyperImpute Jarrett et al. (2022)), and graph neural network-based approaches (MIRACLE Kyono et al. (2021), GRAPE You et al. (2020)). Diffputer demonstrates better performance than these baselines, particularly highlighting its suitability for large data sets ($n > 10,000$). Our results suggest that the performance of Diffputer may not generalize well across different data sets, missing value rates, and types, unlike our proposed imputation-free representation learning method.

The proposed method has several advantages over existing methods for handling missing values. First, an imputation method effective for MCAR-type data may not be optimal for MNAR-type data or data with natural missing values. Imputation errors and data quality compromise can be avoided by allowing for imputation-free representation learning of data with missing values. Deep representation learning of observed values without imputing missing values is valuable in medical research, where synthetic values generated by models are often considered unreliable and can compromise the credibility of data-driven outcomes Zhou et al. (2023). Second, tabular data imputation with mixed data types (e.g., numerical, categorical) requires separate regression and classification models for imputations and train-test data splits for the same process. The proposed method is free from the computational steps and data splits required for imputation. Imputation methods learn missing values from their relationship with observed values. However, as the missing rate increases, the estimation of missing values from fewer observed values impacts imputation accuracy and data quality. The impact of a high missing rate is more pronounced for data with limited samples when fewer observed values are corrupted by the presence of imputed values with high errors. Therefore, recent studies often evaluate imputation methods up to a 30% missing rate Lee & Kim (2023); Wu et al. (2020). Our proposed method mitigates the impact of a higher imputation error in the data resulting from higher missing value rates.

## 7.1 Limitations

Although deep representation learning without imputation or completion of missing values is a significant contribution, the proposed method has several limitations. Classification performance on an imputed data set or data with missing values is an indirect measure of data quality, which is impacted by the sample size. When the missing value rate is low ($< 10\%$) and completely random, imputing missing values can result in better performance than our proposed imputation-free method. Several traditional imputation and classification models are computationally more efficient than the proposed imputation-free classification. The performance gain of the proposed method over other methods diminishes when the sample size is very large, particularly at a low missing rate.

## 8 Conclusions

This paper introduces IFIAL, a novel imputation-free representation learning of tabular data with missing values. The combination of attention mask and incremental learning using feature partitions in IFIAL is computationally effective, performance-wise superior, and robust to missing value rates and types. The proposed IFIAL algorithm outperforms state-of-the-art deep imputation methods in downstream classification, offering the benefits of retaining data quality and efficiently learning high-dimensional feature space incrementally. Future work should aim at investigating imputation-free solutions for other deep learning methods.

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

# A APPENDIX

## A.1 SUMMARY OF TABULAR DATA SETS

Table 4: Summary of the benchmark tabular data sets with heterogeneous data types and structures.

| Type | OpenML Id | Data set | Samples | Features | Numerical | Categorical | Classes |
|---|---|---|---|---|---|---|---|
| | 13 | Breast-cancer | 286 | 9 | 0 | 9 | 2 |
| | 31 | Credit-g | 1000 | 20 | 7 | 13 | 2 |
| | 37 | Diabetes | 768 | 8 | 8 | 0 | 2 |
| | 181 | Yeast | 1484 | 9 | 9 | 0 | 10 |
| Complete | 334 | Monks-problems-2 | 601 | 7 | 0 | 7 | 2 |
| Data sets | 463 | Backache | 180 | 31 | 5 | 26 | 2 |
| (Missing | 1063 | Kc2 | 522 | 21 | 21 | 0 | 2 |
| Values | 1067 | Kc1 | 2109 | 21 | 21 | 0 | 2 |
| Simulated) | 1071 | Mw1 | 403 | 37 | 37 | 0 | 2 |
| | 1480 | Ilpd | 583 | 10 | 9 | 1 | 2 |
| | 1498 | SA-heart | 462 | 9 | 8 | 1 | 2 |
| | 23381 | Dresses-sales | 500 | 12 | 1 | 11 | 2 |
| | 40691 | Wine-quality-red | 1599 | 11 | 11 | 0 | 6 |
| Natural | 55 | Hepatitis | 155 | 19 | 6 | 13 | 2 |
| Missing | 6332 | Cylinder-bands | 540 | 39 | 18 | 21 | 2 |
| Values | 41162 | Kick | 72983 | 10 | 4 | 6 | 2 |
| | 41440 | Okcupid-stem | 50789 | 10 | 9 | 1 | 3 |

## A.2 NUMBER OF OPERATIONS AND COMPUTATIONAL COST

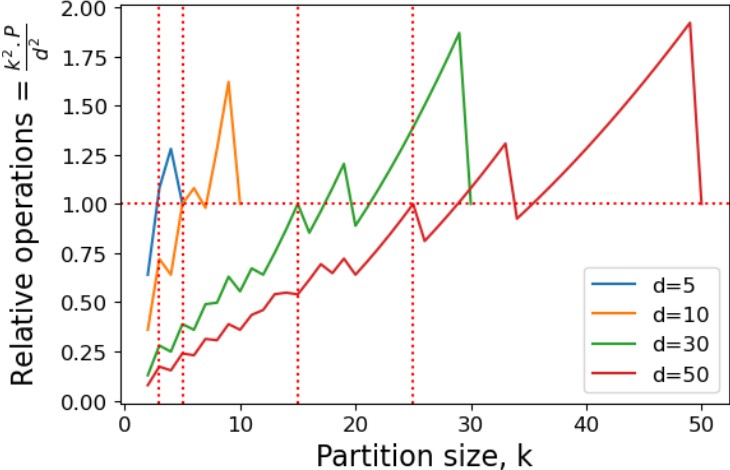

Figure 3: Effects of feature partition size (k) on the computational cost of incremental learning compared to learning the complete feature space (d) at a time. The vertical dotted lines mark the feature partition size when its computational cost equals that of learning the complete feature space or the relative operation ratio is 1.0).

### A.3 EFFECT OF PARTITION SIZE

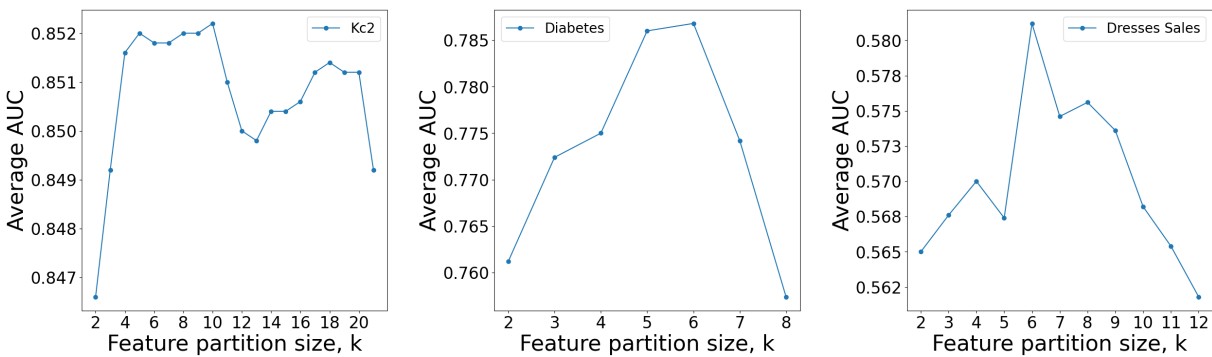

Figure 4: Effects of partition size (k) on average AUC scores obtained across varying missing value rates. The total number of features ($d$) for three data sets are: the Kc2 data set ($d = 21$), Diabetes data set ($d = 8$), and Dresses Sales data set ($d = 12$).

### A.4 COMPUTATIONAL TIME AND ROBUSTNESS TO MISSING RATES

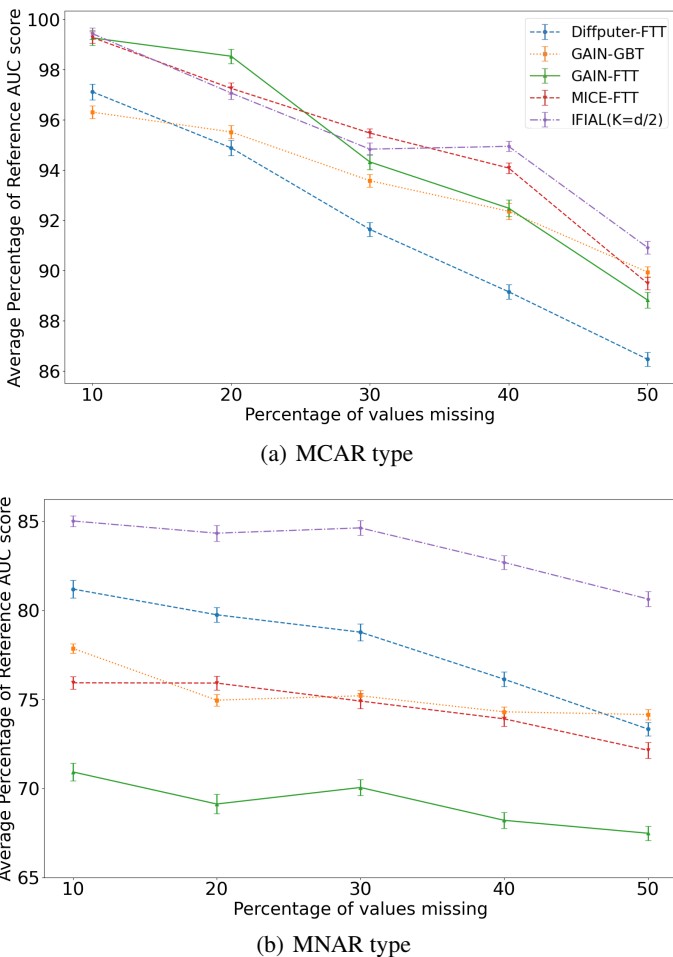

Figure 5: Effect of increasing missing value rates. The average percentage of the reference AUC score is obtained across 13 data sets.

Table 5: Computational runtime in seconds for imputation (training and imputing) and classification (training and inference) steps using the credit-g data set with 50% MCAR type missing values.

| Method | Imputation runtime | Classification runtime | Total runtime | Classification Rank (MCAR) | Classification Rank (MNAR) |
|---|---|---|---|---|---|
| MICE - GBT | 417 | 2 | 419 | 6 | 7 |
| MICE - FTT | 417 | 190 | 607 | 5 | 13 |
| GAIN - GBT | 51 | 2 | 54 | 8 | 11 |
| GAIN - FTT | 51 | 190 | 241 | 3 | 15 |
| Diffputer - GBT | 8 | 1 | 9 | 13 | 14 |
| Diffputer - FTT | 8 | 190 | 198 | 12 | 10 |
| AM-FTT | N/A | 21 | 21 | 15 | 12 |
| MIA - Xgboost | N/A | 1 | 1 | 10 | 6 |
| MIA - LightGBM | N/A | 1 | 1 | 9 | 8 |
| IFIAL ($k = \frac{d}{2}$) | N/A | 163 | 163 | 1 | 1 |

## A.5 DATA SET-SPECIFIC AUC CLASSIFICATION SCORES

Table 6: AUC value for MCAR missing type for two missing rates: 10% and 50%. Rows are data sets, columns are the models.

| Dataset ID | % | Median-GBT | Median-FTT | MICE-GBT | MICE-FTT | GAIN-GBT | GAIN-FTT | Diffputer-FTT | Diffputer-GBT | MIA-Xgboost | MIA-LightGBM | AM-FTT | IFIAL (K=d/2) |
|---|---|---|---|---|---|---|---|---|---|---|---|---|---|
| 1063 | 10 | 0.840 (0.06) | 0.840 (0.05) | 0.837 (0.04) | 0.838 (0.04) | 0.795 (0.04) | 0.846 (0.04) | 0.847 (0.04) | 0.820 (0.05) | 0.791 (0.05) | 0.795 (0.07) | 0.856 (0.04) | 0.862 (0.04) |
| | 50 | 0.802 (0.05) | 0.732 (0.02) | 0.824 (0.06) | 0.833 (0.04) | 0.772 (0.06) | 0.836 (0.05) | 0.849 (0.05) | 0.785 (0.08) | 0.780 (0.05) | 0.773 (0.05) | 0.860 (0.04) | 0.854 (0.04) |
| 463 | 10 | 0.642 (0.16) | 0.717 (0.21) | 0.653 (0.06) | 0.694 (0.09) | 0.672 (0.11) | 0.617 (0.24) | 0.566 (0.21) | 0.654 (0.12) | 0.634 (0.08) | 0.630 (0.06) | 0.587 (0.25) | 0.728 (0.05) |
| | 50 | 0.664 (0.09) | 0.621 (0.21) | 0.565 (0.09) | 0.714 (0.11) | 0.601 (0.03) | 0.596 (0.27) | 0.481 (0.18) | 0.593 (0.13) | 0.583 (0.13) | 0.501 (0.11) | 0.523 (0.14) | 0.674 (0.10) |
| 13 | 10 | 0.713 (0.07) | 0.505 (0.10) | 0.713 (0.07) | 0.691 (0.05) | 0.659 (0.07) | 0.704 (0.06) | 0.647 (0.07) | 0.670 (0.07) | 0.628 (0.08) | 0.643 (0.04) | 0.661 (0.07) | 0.660 (0.07) |
| | 50 | 0.672 (0.09) | 0.489 (0.08) | 0.672 (0.09) | 0.623 (0.04) | 0.628 (0.05) | 0.652 (0.07) | 0.561 (0.08) | 0.628 (0.05) | 0.588 (0.10) | 0.629 (0.06) | 0.488 (0.10) | 0.571 (0.11) |
| 31 | 10 | 0.761 (0.03) | 0.747 (0.03) | 0.749 (0.04) | 0.756 (0.03) | 0.737 (0.03) | 0.733 (0.02) | 0.761 (0.04) | 0.743 (0.02) | 0.743 (0.03) | 0.680 (0.04) | 0.746 (0.03) | 0.753 (0.03) |
| | 50 | 0.657 (0.03) | 0.658 (0.06) | 0.646 (0.04) | 0.704 (0.04) | 0.705 (0.04) | 0.693 (0.05) | 0.695 (0.03) | 0.657 (0.03) | 0.666 (0.05) | 0.642 (0.03) | 0.656 (0.05) | 0.674 (0.04) |
| 37 | 10 | 0.836 (0.03) | 0.841 (0.04) | 0.841 (0.03) | 0.772 (0.04) | 0.794 (0.02) | 0.803 (0.03) | 0.804 (0.03) | 0.805 (0.03) | 0.812 (0.03) | 0.823 (0.03) | 0.804 (0.03) | 0.804 (0.03) |
| | 50 | 0.754 (0.05) | 0.763 (0.06) | 0.724 (0.05) | 0.687 (0.05) | 0.737 (0.02) | 0.548 (0.10) | 0.683 (0.06) | 0.696 (0.07) | 0.706 (0.04) | 0.716 (0.04) | 0.703 (0.04) | 0.718 (0.05) |
| 23381 | 10 | 0.577 (0.05) | 0.580 (0.03) | 0.588 (0.05) | 0.582 (0.03) | 0.621 (0.04) | 0.576 (0.09) | 0.593 (0.06) | 0.570 (0.05) | 0.530 (0.07) | 0.557 (0.05) | 0.603 (0.02) | 0.623 (0.03) |
| | 50 | 0.523 (0.06) | 0.488 (0.10) | 0.520 (0.06) | 0.488 (0.10) | 0.527 (0.05) | 0.561 (0.04) | 0.557 (0.05) | 0.560 (0.09) | 0.509 (0.03) | 0.569 (0.04) | 0.476 (0.06) | 0.516 (0.03) |
| 1480 | 10 | 0.727 (0.03) | 0.708 (0.04) | 0.734 (0.05) | 0.742 (0.04) | 0.731 (0.02) | 0.724 (0.05) | 0.723 (0.07) | 0.733 (0.03) | 0.720 (0.05) | 0.738 (0.05) | 0.663 (0.04) | 0.729 (0.05) |
| | 50 | 0.715 (0.07) | 0.620 (0.09) | 0.684 (0.07) | 0.696 (0.02) | 0.679 (0.03) | 0.650 (0.05) | 0.641 (0.06) | 0.680 (0.03) | 0.665 (0.04) | 0.669 (0.05) | 0.596 (0.06) | 0.678 (0.06) |
| 1067 | 10 | 0.800 (0.02) | 0.791 (0.02) | 0.810 (0.02) | 0.793 (0.02) | 0.799 (0.02) | 0.796 (0.02) | 0.807 (0.02) | 0.806 (0.03) | 0.992 (0.01) | 0.993 (0.00) | 0.797 (0.02) | 0.875 (0.16) |
| | 50 | 0.778 (0.02) | 0.695 (0.03) | 0.792 (0.03) | 0.781 (0.02) | 0.784 (0.03) | 0.788 (0.01) | 0.785 (0.02) | 0.771 (0.03) | 0.796 (0.04) | 0.801 (0.03) | 0.724 (0.02) | 0.793 (0.02) |
| 334 | 10 | 0.684 (0.07) | 0.721 (0.13) | 0.684 (0.07) | 0.805 (0.12) | 0.628 (0.09) | 0.856 (0.01) | 0.721 (0.13) | 0.717 (0.08) | 0.803 (0.04) | 0.756 (0.06) | 0.822 (0.11) | 0.853 (0.08) |
| | 50 | 0.528 (0.06) | 0.531 (0.04) | 0.528 (0.06) | 0.565 (0.04) | 0.537 (0.09) | 0.577 (0.07) | 0.531 (0.04) | 0.532 (0.04) | 0.603 (0.01) | 0.620 (0.04) | 0.553 (0.02) | 0.598 (0.07) |
| 1071 | 10 | 0.650 (0.10) | 0.682 (0.19) | 0.778 (0.07) | 0.778 (0.08) | 0.657 (0.11) | 0.795 (0.06) | 0.734 (0.10) | 0.710 (0.12) | 0.762 (0.05) | 0.760 (0.03) | 0.712 (0.15) | 0.798 (0.03) |
| | 50 | 0.695 (0.06) | 0.665 (0.21) | 0.690 (0.08) | 0.746 (0.11) | 0.685 (0.07) | 0.767 (0.09) | 0.741 (0.09) | 0.650 (0.07) | 0.650 (0.09) | 0.676 (0.08) | 0.667 (0.15) | 0.742 (0.06) |
| 1498 | 10 | 0.700 (0.03) | 0.755 (0.03) | 0.701 (0.03) | 0.744 (0.04) | 0.698 (0.06) | 0.745 (0.05) | 0.738 (0.03) | 0.704 (0.05) | 0.671 (0.04) | 0.675 (0.03) | 0.733 (0.04) | 0.738 (0.04) |
| | 50 | 0.656 (0.03) | 0.693 (0.03) | 0.668 (0.05) | 0.731 (0.04) | 0.683 (0.03) | 0.707 (0.03) | 0.649 (0.06) | 0.684 (0.06) | 0.631 (0.06) | 0.641 (0.05) | 0.654 (0.09) | 0.652 (0.08) |
| 40691 | 10 | 0.770 (0.03) | 0.746 (0.01) | 0.789 (0.03) | 0.704 (0.04) | 0.758 (0.05) | 0.735 (0.02) | 0.733 (0.02) | 0.736 (0.05) | 0.834 (0.02) | 0.829 (0.02) | 0.724 (0.02) | 0.744 (0.02) |
| | 50 | 0.692 (0.04) | 0.669 (0.01) | 0.685 (0.04) | 0.644 (0.02) | 0.698 (0.01) | 0.682 (0.02) | 0.604 (0.02) | 0.604 (0.05) | 0.699 (0.03) | 0.718 (0.04) | 0.654 (0.02) | 0.664 (0.01) |
| 181 | 10 | 0.819 (0.04) | 0.792 (0.02) | 0.798 (0.05) | 0.744 (0.02) | 0.803 (0.03) | 0.808 (0.02) | 0.784 (0.02) | 0.803 (0.04) | 0.832 (0.03) | 0.835 (0.03) | 0.738 (0.01) | 0.768 (0.01) |
| | 50 | 0.686 (0.06) | 0.716 (0.02) | 0.701 (0.06) | 0.642 (0.02) | 0.710 (0.05) | 0.505 (0.02) | 0.634 (0.02) | 0.698 (0.06) | 0.731 (0.05) | 0.724 (0.05) | 0.642 (0.02) | 0.704 (0.01) |

Table 7: AUC value for MNAR missing type for two missing rates: 10% and 50%. Rows are data sets, columns are the models.

| Dataset ID | % | Median-GBT | Median-FTT | MICE-GBT | MICE-FTT | GAIN-GBT | GAIN-FTT | Diffputer-GBT | Diffputer-FTT | MIA-Xgboost | MIA-LightGBM | AM-FTT | IFIAL (K=d/2) |
|---|---|---|---|---|---|---|---|---|---|---|---|---|---|
| 1063 | 10 | 0.892 (0.05) | 0.623 (0.11) | 0.863 (0.04) | 0.668 (0.14) | 0.496 (0.03) | 0.509 (0.02) | 0.468 (0.06) | 0.525 (0.11) | 0.904 (0.04) | 0.912 (0.04) | 0.783 (0.07) | 0.871 (0.05) |
|  | 50 | 0.553 (0.06) | 0.536 (0.05) | 0.506 (0.04) | 0.499 (0.05) | 0.505 (0.02) | 0.512 (0.03) | 0.507 (0.08) | 0.486 (0.05) | 0.540 (0.07) | 0.542 (0.08) | 0.492 (0.04) | 0.557 (0.03) |
| 463 | 10 | 0.627 (0.09) | 0.675 (0.06) | 0.596 (0.11) | 0.485 (0.12) | 0.672 (0.10) | 0.627 (0.22) | 0.532 (0.08) | 0.541 (0.21) | 0.665 (0.08) | 0.595 (0.10) | 0.406 (0.09) | 0.746 (0.06) |
|  | 50 | 0.652 (0.13) | 0.595 (0.12) | 0.618 (0.14) | 0.439 (0.12) | 0.557 (0.09) | 0.595 (0.15) | 0.663 (0.09) | 0.563 (0.11) | 0.623 (0.10) | 0.543 (0.11) | 0.602 (0.21) | 0.657 (0.10) |
| 13 | 10 | 0.704 (0.08) | 0.657 (0.08) | 0.704 (0.08) | 0.645 (0.06) | 0.701 (0.09) | 0.527 (0.13) | 0.666 (0.08) | 0.694 (0.06) | 0.698 (0.08) | 0.655 (0.06) | 0.657 (0.08) | 0.676 (0.07) |
|  | 50 | 0.642 (0.09) | 0.583 (0.08) | 0.642 (0.09) | 0.630 (0.11) | 0.676 (0.10) | 0.487 (0.16) | 0.621 (0.07) | 0.674 (0.09) | 0.602 (0.10) | 0.638 (0.10) | 0.583 (0.08) | 0.607 (0.10) |
| 31 | 10 | 0.696 (0.01) | 0.745 (0.04) | 0.690 (0.01) | 0.534 (0.06) | 0.730 (0.03) | 0.537 (0.11) | 0.710 (0.04) | 0.739 (0.04) | 0.693 (0.03) | 0.680 (0.04) | 0.494 (0.02) | 0.742 (0.03) |
|  | 50 | 0.637 (0.02) | 0.679 (0.01) | 0.629 (0.03) | 0.540 (0.06) | 0.657 (0.03) | 0.507 (0.02) | 0.604 (0.06) | 0.643 (0.04) | 0.661 (0.03) | 0.642 (0.03) | 0.531 (0.03) | 0.684 (0.02) |
| 37 | 10 | 0.510 (0.02) | 0.525 (0.03) | 0.491 (0.05) | 0.524 (0.04) | 0.497 (0.01) | 0.502 (0.01) | 0.554 (0.05) | 0.522 (0.02) | 0.540 (0.02) | 0.530 (0.02) | 0.498 (0.03) | 0.503 (0.04) |
|  | 50 | 0.504 (0.05) | 0.489 (0.07) | 0.527 (0.05) | 0.510 (0.04) | 0.502 (0.02) | 0.500 (0.01) | 0.537 (0.05) | 0.488 (0.05) | 0.502 (0.04) | 0.526 (0.03) | 0.494 (0.04) | 0.502 (0.04) |
| 23381 | 10 | 0.570 (0.06) | 0.625 (0.06) | 0.570 (0.06) | 0.486 (0.02) | 0.557 (0.04) | 0.538 (0.08) | 0.531 (0.16) | 0.621 (0.06) | 0.555 (0.05) | 0.582 (0.05) | 0.557 (0.10) | 0.615 (0.04) |
|  | 50 | 0.581 (0.06) | 0.573 (0.05) | 0.586 (0.07) | 0.506 (0.01) | 0.566 (0.13) | 0.452 (0.11) | 0.435 (0.15) | 0.543 (0.02) | 0.599 (0.03) | 0.499 (0.04) | 0.551 (0.06) | 0.566 (0.05) |
| 1480 | 10 | 0.542 (0.05) | 0.534 (0.04) | 0.539 (0.04) | 0.520 (0.04) | 0.547 (0.02) | 0.511 (0.05) | 0.581 (0.05) | 0.534 (0.03) | 0.558 (0.03) | 0.551 (0.04) | 0.518 (0.07) | 0.559 (0.06) |
|  | 50 | 0.537 (0.07) | 0.543 (0.04) | 0.531 (0.03) | 0.528 (0.04) | 0.556 (0.03) | 0.479 (0.04) | 0.546 (0.05) | 0.530 (0.06) | 0.519 (0.08) | 0.523 (0.08) | 0.535 (0.04) | 0.546 (0.01) |
| 1067 | 10 | 0.802 (0.02) | 0.690 (0.04) | 0.987 (0.01) | 0.957 (0.01) | 0.503 (0.01) | 0.462 (0.09) | 0.486 (0.01) | 0.798 (0.11) | 0.992 (0.01) | 0.993 (0.01) | 0.978 (0.01) | 0.880 (0.04) |
|  | 50 | 0.701 (0.08) | 0.600 (0.02) | 0.796 (0.03) | 0.732 (0.06) | 0.510 (0.01) | 0.550 (0.09) | 0.484 (0.03) | 0.568 (0.04) | 0.796 (0.04) | 0.801 (0.03) | 0.792 (0.02) | 0.647 (0.09) |
| 334 | 10 | 0.643 (0.05) | 0.806 (0.10) | 0.643 (0.05) | 0.767 (0.11) | 0.588 (0.05) | 0.499 (0.04) | 0.622 (0.06) | 0.742 (0.17) | 0.811 (0.04) | 0.767 (0.04) | 0.806 (0.10) | 0.843 (0.02) |
|  | 50 | 0.510 (0.01) | 0.575 (0.05) | 0.510 (0.01) | 0.537 (0.05) | 0.504 (0.06) | 0.481 (0.03) | 0.495 (0.04) | 0.510 (0.04) | 0.578 (0.02) | 0.553 (0.05) | 0.575 (0.05) | 0.621 (0.07) |
| 1071 | 10 | 0.584 (0.08) | 0.462 (0.10) | 0.526 (0.06) | 0.464 (0.10) | 0.525 (0.03) | 0.483 (0.03) | 0.470 (0.04) | 0.526 (0.09) | 0.536 (0.07) | 0.546 (0.07) | 0.466 (0.12) | 0.430 (0.05) |
|  | 50 | 0.612 (0.03) | 0.421 (0.11) | 0.547 (0.12) | 0.442 (0.12) | 0.508 (0.01) | 0.503 (0.01) | 0.448 (0.08) | 0.404 (0.06) | 0.468 (0.11) | 0.445 (0.10) | 0.463 (0.07) | 0.429 (0.16) |
| 1498 | 10 | 0.616 (0.09) | 0.633 (0.05) | 0.593 (0.09) | 0.484 (0.03) | 0.632 (0.06) | 0.614 (0.10) | 0.571 (0.05) | 0.617 (0.06) | 0.546 (0.05) | 0.572 (0.06) | 0.512 (0.04) | 0.642 (0.06) |
|  | 50 | 0.569 (0.06) | 0.598 (0.05) | 0.556 (0.03) | 0.431 (0.06) | 0.577 (0.02) | 0.457 (0.06) | 0.576 (0.05) | 0.587 (0.03) | 0.536 (0.09) | 0.530 (0.08) | 0.543 (0.07) | 0.602 (0.04) |
| 40691 | 10 | 0.555 (0.04) | 0.486 (0.03) | 0.488 (0.06) | 0.497 (0.02) | 0.502 (0.03) | 0.497 (0.01) | 0.500 (0.02) | 0.499 (0.03) | 0.537 (0.04) | 0.539 (0.05) | 0.713 (0.01) | 0.478 (0.04) |
|  | 50 | 0.512 (0.05) | 0.506 (0.02) | 0.511 (0.01) | 0.505 (0.03) | 0.500 (0.02) | 0.487 (0.01) | 0.490 (0.03) | 0.499 (0.03) | 0.474 (0.06) | 0.490 (0.05) | 0.642 (0.02) | 0.494 (0.03) |
| 181 | 10 | 0.524 (0.03) | 0.514 (0.01) | 0.531 (0.03) | 0.521 (0.02) | 0.502 (0.04) | 0.505 (0.02) | 0.510 (0.05) | 0.514 (0.01) | 0.564 (0.03) | 0.571 (0.04) | 0.575 (0.01) | 0.563 (0.02) |
|  | 50 | 0.473 (0.02) | 0.507 (0.02) | 0.493 (0.02) | 0.507 (0.02) | 0.502 (0.05) | 0.499 (0.01) | 0.505 (0.03) | 0.496 (0.02) | 0.528 (0.03) | 0.521 (0.02) | 0.522 (0.01) | 0.511 (0.02) |

