# OpenReview forum: "Imputation-free Learning of Tabular Data with Missing Values using Incremental Feature Partitions in Transformer"
_ICLR.cc/2026/Conference — Submitted to ICLR 2026_

### Official Review · Reviewer_UaUB · 2025-11-01

**Soundness:** 2
**Presentation:** 2
**Contribution:** 2
**Rating:** 2
**Confidence:** 3

**Summary:**

This paper proposes a representation learning method for incomplete, heterogeneous tabular data (with both numerical and categorical variables) based on a transformer called Imputation-Free Incremental Attention Learning (IFIAL). The idea is to learn representation using missing masks directly through attention and incremental learning using feature partitions. The empirical evaluation on 17 tabular datasets from the OpenML repository shows that IFIAL produced better results and is robust to missing rate and mechanism (MCAR and MNAR) when compared to methods proposed to handle missing data based on imputation.

**Strengths:**

1. The paper tackles a fundamental issue of handling missing data, which is very common in practical data analysis and machine learning problems, where some data are missing due to reasons like sensor malfunction, communication problems, or privacy concerns.

2. It attempts to address issues related to a significant missing rate (>10%), different missing mechanisms (MCAR and MNAR) and heterogeneous data.

3. Performs a lot of experiments on 17 datasets, though most of them are small.

**Weaknesses:**

1. Representation-learning-based methods to learn complete latent representation from incomplete data (e.g., kernel-based and PCA-based methods) are not discussed and used for comparison, though they mentioned this approach in the introduction. I believe these methods are direct competitors of the proposed method. Some examples of related literature that are worth looking at include:
L. A.Belanche, V. Kobayashi, and T. Aluja. Handling missing values in kernel methods with application to microbiology data. Neurocomputing, 141:110–116, 2014.
F. M. Bianchi, L. Livi, K. Ø. Mikalsen, M. Kampffmeyer, and R. Jenssen. Learning representations of multivariate time series with missing data. Pattern Recognition, 96:106973, 2019.
G. Sanguinetti and N. D. Lawrence. Missing data in kernel pca. In European Conference on Machine Learning, pages 751–758. Springer, 2006.
M. ´ Smieja, Ł. Struski, J. Tabor, B. Zieli´nski, and P. Spurek. Processing of missing data by neural networks. Advances in neural information processing systems, 31, 2018.
M. ´ Smieja, Ł. Struski, J. Tabor, and M. Marzec. Generalized rbf kernel for incomplete data. Knowledge-Based Systems, 173:150–162, 2019.
A. J. Smola, S. V. N. Vishwanathan, and T. Hofmann. Kernel methods for missing variables. In Proceedings of the Tenth International Workshop on Artificial Intelligence and Statistics, pages 325–332, 2005.
F. Yu, R. Zhao, Z. Shi, Y. Lu, J. Fan, Y. Zeng, J. Mao, and W. Li. Boosting spectral clustering on incomplete data via kernel correction and affinity learning. Advances in Neural Information Processing Systems, 36:72583–72603, 2023.
F. Yu, Y. Zeng, J. Mao, and W. Li. A theory-driven approach to inner product matrix estimation for incomplete data: An eigenvalue perspective. In Proceedings of the ACM on Web Conference 2025, pages 4077–4088, 2025.

* The paper lacks a clear intuition/motivation for why the proposed approach works.

* The proposed method is not explained clearly; some key details on how IFAIL works are missing. For example, there are some inconsistencies with notations (e.g., n and N_1, N_2 to represent the number of data points). It looks like some notations are not clearly defined (e.g., X_i). What is the rationale of using Eqn 1 to determine the number of partitions? It looks like M(\theta) is reset instead of updated with each P_i (according to Algo 1), etc.

* Empirical evaluation is slim - used very small and relatively low-dimensional datasets - only two out of 17 datasets have more than 2100 samples, and dimensionality ranges from 7 to 39. The authors themselves said that deep-learning-based methods are not good in tabular datasets due to the lack of sufficient training samples.

**Questions:**

Please see the weaknesses raised above. I am particularly interested in authors' responses to the following:
* Why are kernel-based representation learning methods not used in comparison?
* How did you simulate the MNAR missing scenario? How do you ensure the missing rate of r% in a dataset? Did you use a fixed r% missing in a certain proportion of features, or is it different between features?
* It looks like the IFIAL requires some features where there are no missing values. Does it work in the case where there is some data missing in all variables?
* You said that a pretrained language model is used to encode feature names and categorical labels. Could you please provide a bit more detail on the model used and conduct a sensitivity analysis using different models?
* In the second paragraph of Section 1, you mention that Section 4 introduces the proposed NAIL strategy. I am not sure if this is a typo or something. I don't NAIL is discussed in Sec 4. Am I missing something here? Would you please clarify it.

---

### Official Review · Reviewer_sFdh · 2025-11-01

**Soundness:** 2
**Presentation:** 1
**Contribution:** 2
**Rating:** 2
**Confidence:** 4

**Summary:**

This paper introduces an imputation-free representation learning of tabular data with missing values. The proposed method employs attention mask and incremental learning with feature partitions. The experiments on 17 datasets demonstrate that the proposed method outperforms state-of-the-art imputation methods in downstream classification tasks.

**Strengths:**

S1. The idea of imputation-free representation learning is interesting.

S2. The proposed method outperforms various state-of-the-art imputation methods on 17 datasets.

S3. The advantages and limitations of the proposed method are comprehensively discussed.

**Weaknesses:**

W1. The presentation needs improvement. The introduction part is way too short, rendering the contributions of this paper unclear.

W2. The novelty is unclear because attention mechanisms have been widely used for imputation (see the following references). Although this paper targets imputation-free incremental attention learning, the two problems are very related, and the imputation performance is often evaluated in these related works using downstream tasks. Moreover, these methods are not evaluated in the experiments of this paper either.

[1] Wu et al. Attention-based learning for missing data imputation in HoloClean. MLSys 2020.
[2] Tihon et al. DAEMA: Denoising autoencoder with mask attention. ICANN 2021.
[3] Kawagoshi et al. CAGAIN: Column attention generative adversarial imputation networks. DEXA 2023.

W3. The proposed method employs the idea of overlapping columns in FTT, further compromising its novelty.

W4. In Tables 1 and 2, average performance ranks are reported instead of AUC scores. This average rank is not a good metric because large values may dominate the result. For example, (1, 1, 1, 10) yields an average rank of 3.25, worse than (3, 3, 3, 3) which has an average rank of 3.

W5. The efficiency claim needs to be revised because Table 5 shows the proposed method ranks 6th among 10 methods compared.

**Questions:**

Q1. Why is the overlap size (s) ceil(k/2)? How does it affect the proposed method's performance?

Q2. It is unclear how y_i is obtained in Algorithm 1. What is the relation between y_i and y?

Q3. In Figure 3, k is evaluated up to 50, and the default value of k is determined to be d/2. So I don't quite understand what "We primarily evaluate the proposed incremental learning method for three partition sizes, k = 2, 3, and 4." means in Sec. 5.2.

---

### Official Review · Reviewer_ejer · 2025-11-03

**Soundness:** 1
**Presentation:** 1
**Contribution:** 2
**Rating:** 0
**Confidence:** 4

**Summary:**

would be great to mention in paragraph 1 that decision-tree based methods can bypass needing to imputation missing values, they can just make split in certain ways when the data is missing. So having a system that can also naturally handle missing data in deep learning would be great.

The related work is one big paragraph, please organize it.

3, motivation: well you are not supposed to impute with both train and test. Indeed it would cause a leak. Normally people train on training data and use the model to impute on test data. I see so these people Du and Lall are cheating...

The algorithm traines with partition of features, from least missing to more missing data. Attention maskes are used to handle missing values.

" Notably, FTT tokenizes features and uses a pre-trained language model to obtain embeddings for feature names and categorical features." Wait, why do you need a LLM? Cant you just learn a nn.Embedding? Is there a real benefit of having embeddings for feature names? And what if the feature as no name, what do you do then? (var1, var2, ...). This seems over-engineered.

"In contrast, numerical values are passed through a linear projection layer to obtain corresponding embeddings." Why only linear and not an MLP. Have you tried an MLP?

Good choices of datasets and missing values.

Would be great to have the SOTA for imputation which is MissForest.

Table should mention in the caption that its 5-fold CV AUC.

Why is there only AUC? Are you not solving regression problems as well? You should have numbers for regression.

Why are you only comparing FTT to GBT? Is FTT the SOTA for tabular data with deep learning? You should have other baselines.

You say GBT, but there are many GBT methods, which one is it? You should show Xgboost, LightGBM with MICE and MissForest as comparision baselines.

Right now the XgBoost and LightGBM numbers seems really bad, this makes me highly suspect unfair play. Are you properly tuning the hyperparameters, trying let say 100 different hyperparameters? See https://arxiv.org/pdf/2209.15421 Appendix E Table 15 for an example of 100 hyperparameters search with CatBoost. Theres no mention of hyperparamters for the baseline actually, this is weird.

See https://arxiv.org/pdf/2402.03970v2 and https://arxiv.org/pdf/2207.08815 as good references for deep learning vs tabular data. You need more of these reference.

Paper is poorly written, it should be rewritten.

Weird to have experiments and results as two sections.

**Strengths:**

many datasets in the experiments (but only classification)

**Weaknesses:**

See "Summary"

**Questions:**

See Summary, I have many.

---

### Official Review · Reviewer_PEQg · 2025-11-03

**Soundness:** 2
**Presentation:** 3
**Contribution:** 2
**Rating:** 2
**Confidence:** 4

**Summary:**

The paper proposes a transformer-based method for deep tabular learning that operates directly in the presence of missing values. The authors demonstrate superior performance on selected benchmark datasets. However, the manuscript does not provide information on how baseline methods were tuned (for example, hyperparameter optimisation), and several key contemporary deep-learning baselines are absent from the comparison.

**Strengths:**

- The motivation is clear. Tabular data frequently contains missing values, and handling such missingness remains a challenging and impactful problem. Imputation strategies can substantially distort feature distributions and downstream predictive performance, which underscores the relevance of an imputation-free approach

- The paper is well written and generally easy to follow

- The authors provide an implementation of the method

**Weaknesses:**

- The manuscript does not sufficiently detail the hyperparameter optimisation strategy for baseline methods. Given the sensitivity of GBDT and deep architectures to hyperparameters, the absence of a transparent search protocol (search space, budget, tuning criteria, and early-stopping) raises concerns regarding the fairness and reproducibility of the benchmarking. I recommend employing a standardized benchmarking framework (for example, TabArena or the protocol from [1]) and reporting the procedure clearly. It would also be informative to evaluate IFIAL on fully observed datasets to contextualize its performance in general tabular learning settings.

- The baseline suite is limited largely to GBDT pipelines and earlier transformer baselines. Recent high-performing deep tabular architectures, particularly transformer-based models such as TabPFN-v2, TabM, TabLLM, and other current models, are absent. Inclusion of modern architectures would strengthen the comparative analysis.

- The evaluation focuses only on AUC. A broader set of metrics (for example, accuracy and F1) would provide a more comprehensive performance assessment, especially for imbalanced datasets.

- The proposed method requires longer runtime and the utilization of a GPU, which makes it less favourable in comparison to GBDT algorithms in settings where computational efficiency is a priority.

**Questions:**

Please see the weaknesses section.

---

### Official Review · Reviewer_m7f9 · 2025-11-04

**Soundness:** 1
**Presentation:** 2
**Contribution:** 1
**Rating:** 2
**Confidence:** 4

**Summary:**

This paper proposes IFIAL, a transformer-based framework to handle tabular data with missing values. However, the problem of missing data is well-studied, and introducing another complex, 'black-box' model is a significant drawback. This is especially true for tabular data in domains that require explainability, where many existing and transparent traditional methods are preferred. Overall, the paper's claimed contribution is not significant enough for a top ML conference, and there are significant issues regarding the experimental design and evaluation of the proposed method.

Please find my detailed comments below.

**Strengths:**

The authors conduct a wide-ranging benchmark, comparing their proposed method against numerous baselines across diverse datasets. The paper also includes ablation studies.

**Weaknesses:**

1. The evaluation focuses only on empirical evaluations using real-world datasets and fails to include simulated settings where the ground-truth data generating process is known. Since many research fields focus on non-prediction tasks (e.g., identifying relations between outcomes and features, causal analysis), simulation studies where ground-truth parameters are known are critical for properly evaluating methods for tabular data.

2. The evaluation is severely limited in scope. It focuses only on classification tasks, providing no results for regression (predicting continuous outcomes). Furthermore, the evaluation is concerned only with downstream predictive performance, with no analysis of how the method impacts non-prediction tasks like feature importance or coefficient recovery, which are critical use cases for tabular data.

3. The studies lack statistical rigor to support the authors' claims. The paper reports results from a single 5-fold cross-validation run (five runs total), which is far from sufficient to establish statistical power or stable conclusions. A rigorous study would require hundreds of replications to reliably compare methods and perform statistical tests.

**Questions:**

Can the authors please add the results suggested by the weaknesses above? To substantiate the paper's claims, a more complete evaluation is necessary. This should at least include: simulation studies using synthetic data where the ground-truth is known; evaluations on regression tasks (predicting continuous outcomes and evaluations of the method's performance on non-prediction tasks, such as the stability or accuracy of feature importance measures or point estimates.

---

### Meta-Review · Area_Chair_cGZX · 2025-12-19

**Summary:**

Tabular data with missing values make learning tasks difficult, where many of existing methods focus on the "impute-then-learn" paradigm. This manuscript works under the setting of no imputation and direct learning from tabular data with missing values. The main idea is to sort the feature columns according to the missing rate and then incrementally group subset of features. These subset of features are then passed to a transformer with masked attention according to the missing value indicator mask. Experiments compared the proposed method with many baselines (including those with our without imputation) and the proposed method is claimed to achieve the best results.

Reviewers are mainly concerned about the experimental settings, e.g., lack of many transformer based baselines, insufficient discussion of the impact of missing mechanism (MCAR, MAR, MNAR) on the effectiveness of the proposed approach, etc.

**Reviewer Concerns:**

Reviewers' concerns are mainly on experiments:
- Unclear details regarding hyper-parameter optimization.
- Baseline methods largely limited to GBDT and earlier transformer baselines, need to add recent Transformer methods e.g., TabPFN.
- Evaluation metric is AUC only and all the datasets are classification datasets.

No author rebuttal is provided.

**Reviewer Scores:**

Reviewers voted for rejection.

No author rebuttal is submitted.

---

### Decision · Program_Chairs · 2026-01-26

Reject